



# Indoor $^{222}$Rn Modeling in Data-Scarce Regions: An Interactive Dashboard Approach for Bogotá, Colombia

Martín Domínguez Durán[1,2], María Angélica Sandoval Garzón[3], and Carme Huguet[1]

[1]Geosciences department, Los Andes University, Bogotá, Colombia
[2]Civil and Environmental Engineering department, Los Andes University, Bogotá, Colombia
[3]Pedagogical and Technological University of Colombia, Tunja, Colombia (UPTC)

**Correspondence:** Carme Huguet (m.huguet@uniandes.edu.co)

**Abstract.**

Radon ($^{222}$Rn) is a naturally occurring gas that represents a health threat due to its causal relationship with lung cancer. Despite its potential health impacts, several regions have not conducted studies, mainly due to data scarcity and/or economic constraints. This study aims to bridge the baseline information gap by building an interactive dashboard that uses inferential statistical methods to estimate indoor radon concentration's (IRC) spatial distribution for a target area. We demonstrate the functionality of the dashboard by modeling IRC in the city of Bogotá, Colombia, using 30 in situ measurements. IRC measured were the highest reported in the country, with a geometric mean of 91 $\pm$14 Bq/m$^3$ and a maximum concentration of 407 Bq/m$^3$. In 57% of the residences RC exceeded the WHO's recommendation of 100 Bq/m$^3$. A prediction map for houses registered in Bogotá's cadaster was built in the dashboard by using a log-linear regression model fitted with the in situ measurements, together with meteorological, geologic and building specific variables. The model showed a cross-validation Root Mean Squared Error of 56.5 $\frac{Bq}{m^3}$. Furthermore, the model showed that the age of the house presented a statistically significant positive association with RC. According to the model, IRC measured in houses built before 1980 present a statistically significant increase of 72% compared to those built after 1980 (p-value = 0.045). The prediction map exhibited higher IRC in older buildings most likely related to cracks in the structure that could enhance gas migration in older houses. This study highlights the importance of expanding $^{222}$Rn studies in countries with a lack of baseline values and provides a cost-effective alternative that could help deal with the scarcity of IRC data and get a better understanding of place-specific variables that affect IRC spatial distribution.

## 1 Introduction

Radon gas ($^{222}_{86}$Rn) represents a natural hazard that can have great impact on human health and remains largely unexplored in some regions (World Health Organization, 2019). This noble gas is colourless, odourless, highly radioactive and is part of the $^{238}$U decay chain (Field, 2015). In 1988, $^{222}$Rn was classified as a group 1 carcinogenic agent (IARC, 1988) because itself and its decay products (i.e. $^{218}$Po and $^{214}$Po) emit alpha particles and, at high $^{222}$Rn concentrations (i.e. above 100 Bq/m$^3$), these particles can affect the lung's epithelial tissues and lead to the development of lung cancer and other medical conditions (Auvinen et al., 2005; Turner et al., 2012; Field, 2015; Lehrer et al., 2017; Ruano-Ravina et al., 2017). The risk of lung cancer



increases 16% for every 100 Bq/m$^3$ increment in residential $^{222}$Rn concentrations (e.g. Darby et al., 2005). In the 66 countries that have had national $^{222}$Rn surveys, it is estimated that 226,057 people die every year from lung cancer due to $^{222}$Rn exposure (Gaskin et al., 2018). This type of estimations has prompted some national and international entities to assess residential $^{222}$Rn levels and implement policies to address this public health issue (WHO, 2009).

Despite radon's proven adverse health effects, there remain evident disparities between countries regarding $^{222}$Rn policies and monitoring programs. In 2007 an international $^{222}$Rn survey was carried out by the World Health Organization (WHO). In it, 32 countries, of which 81% are part of the global north, presented their residential radon mean values (WHO, 2007). Twelve years later, the global health observatory show that the number of countries that have reported their average indoor radon levels had risen to 44. However, only 22% of these countries are in the global south (See Fig. 1a). Furthermore, only one out of

the 24 countries committed to evaluate IRC using a national radon database is in the global south (See Fig. 1b). For multiple countries, especially in Latin America, Africa and Asia, discrete studies, in which mean levels were calculated for one or two cities, were used to report national levels. Nevertheless, as radon presents high spatial variability, this type of discrete studies cannot represent an entire country.

In terms of action levels, even though some organizations like the WHO and the United States Environmental Protection Agency (USEPA) recommend IRC to remain under 100 Bq/m$^3$ and 148 Bq/m$^3$ respectively, multiple countries still do not have any action level or their levels don't align with the ones recommended (USEPA, 1987; WHO, 2009). In South and Central America, only 7 countries have governmental regulations in which action levels are presented. Notably, countries like Argentina or Chile do not have any indoor radon regulations. Additionally, the action levels in the countries with regulations largely surpass the recommended values (Giraldo-Osorio et al., 2020). For instance, in the case of Colombia, the action level

is 400 Bq/m$^3$ and there is no $^{222}$Rn monitoring program (Ministerio de Minas y Energía, 2002).

While efforts have been made to assess this public health issue in Latin America and the Caribbean (LAC), IRC remain largely unexplored in the region (Canoba et al., 2002; Giraldo-Osorio et al., 2020). For instance, in Colombia only two resi-

dential $^{222}$Rn studies have been conducted so far, both in the urban and rural areas of Manizales with 18 and 202 dwellings measured respectively (Garzon et al., 2013; Giraldo-Osorio et al., 2021). There is thus a clear lack of baseline information about this carcinogenic gas in the country as well as the LAC region. This lack of baseline information may be a reason why regulations regarding $^{222}$Rn are outdated or still missing in the region (Giraldo-Osorio et al., 2020). It is paramount to find innovative ways to bridge this information gap more efficiently. In this study, the potential of dashboards to give place specific

estimates of IRC is presented.



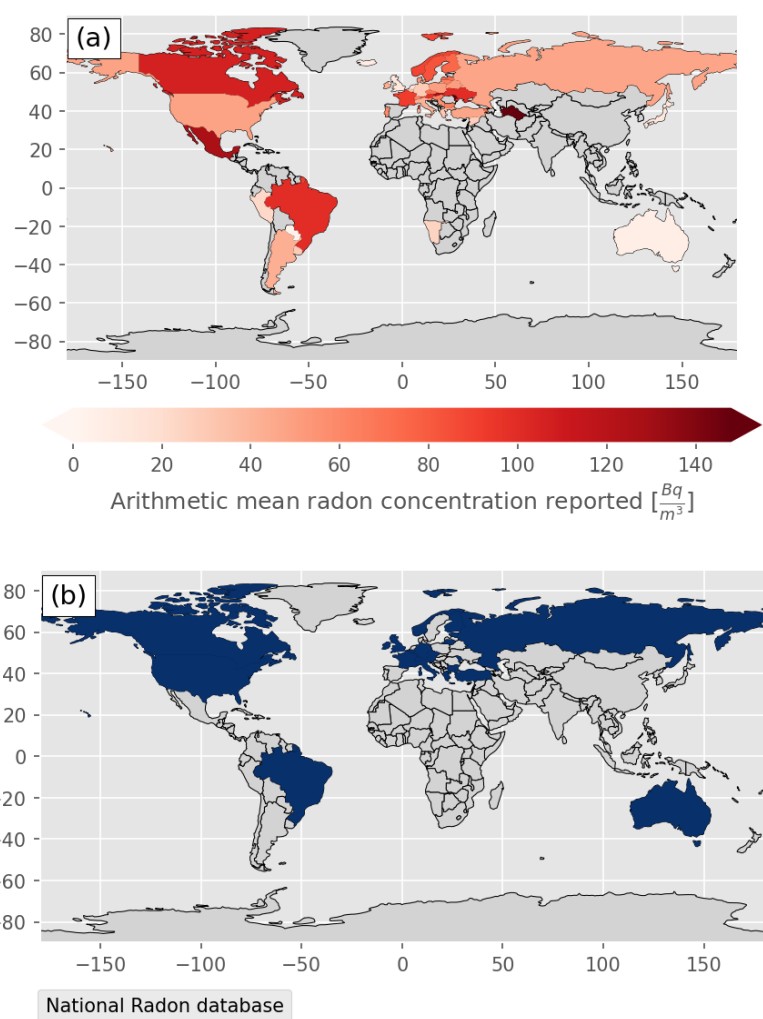

**Figure 1.** Spatial distribution of indoor radon measurements by country. Countries coloured in grey do not have reported any radon information. **(a)** Arithmetic mean values reported per country $\left[\frac{Bq}{m^3}\right]$. **(b)** Countries with an existing radon database are coloured in blue. Data retrieved from World Health Organization (2019). Figure created by authors.

Dashboards are organized information systems whose main objective is to summarize quantitative data in a way that can be easily understood and used by a target audience (Nijkamp and Kourtit, 2022). In public health, dashboards may guide agency decisions on resource allocation (Dasgupta and Kapadia, 2022). The dashboard created in this study aims to: Understand the variables that can affect the radon distribution for a specific place, provide summarized IRC information for place-specific policy and decision making, guide the direction in which further studies should be made, and be a potential tool to estimate IRC distribution in cities where IRC measurements are scarce. To the best of the authors' knowledge this is the first dashboard developed for performing radon modeling.



The high spatial variability of radon is well known and has been associated with three main types of variables that could act as predictors of IRC: meteorological, geologic and building specific variables (e.g. (Gundersen et al., 1993; Karpińska et al., 2009; Mullerova et al., 2017, etc.)). These variables can either increase or decrease residential IRC levels and their influence depends greatly on the location. For example, a city's IRC might be highly dependant on building specific variables while another city's IRC might be more influenced by geologic variables. Understanding the factors associated with IRC distribution in a city is crucial to assess the magnitude of a possible public health hazard.

Several models have been used to estimate radon concentrations in residences previously (Demoury et al., 2013; Elío et al., 2017; Vienneau et al., 2021). These models can be either mechanistic or statistical. Some advantages of statistical models are: they quantify the relationship between variables and they allow the inclusion of different types of variables, even if the underlying association between them are not yet well understood. On the other hand, mechanistic models are based on the understanding of the physical mechanisms that control a phenomenon. For instance, in the case of indoor Air Quality, mechanistic models make sure to understand the mechanisms governing the transport of the pollutant (Wei et al., 2019). The dashboard designed in this study uses a log-linear regression model to recognize the most influential variables and estimate IRC by using these variables as predictors.

Eventhough Bogotá's geologic, climatic and construction characteristics could facilitate high IRC, no baseline studies have been conducted in the city. This makes Bogotá an ideal place to evaluate the performance of the dashboard in estimating the IRC spatial distribution in a data-scarce region by using statistical learning.

The structure of this paper is as follows. The methodology is presented in section 2, where the study area and the data acquisition is presented first (subsection 2.1), and then the statistical analysis and the dashboard architecture is explained in subsection 2.2. Next, the results for the city of Bogotá are presented and discussed in section 3. First, the indoor radon levels detected in situ are presented in subsection 3.1. Then, in subsections 3.2 and 3.3 the results of the modeling dashboard are discussed in terms of the potential influential factors for high IRC and the potential hazard posed by $^{222}$Rn in Bogotá. Next, the dashboard's functionality was assessed in subsection 3.4.

## 2 Materials and methods

### 2.1 Study area and Data acquisition

#### 2.1.1 Study area

Bogotá is located at the western side of the eastern Andes cordillera at an average altitude of 2640 m above sea level (Fig. 2). The urban area of Bogotá is divided in 20 localities, 5 of which were included in the present study (Fig. 2, Departamento



Administrativo Nacional de Estadística, 2018b). Bogota's temperature ranges between 8°C and 19°C throughout the year with an annual average value of 13°C. It has an average precipitation of 80 $\frac{mm}{month}$ throughout the year with a minimum around 40 mm in January and a maximum around 120 mm in May (IDEAM, 2015). The geology of the Bogotá region is characterized by outcrops from the late Cretaceous to the Quaternary (Montoya and Reyes, 2007). According to the Geologic map of Colombia (Gómez Tapias et al., 2015) the main lithologies in Bogotá with their respective chronostratigraphic codes are: limestones and

shales (k1k6-Stm) near La Calera municipality, alluvial fans and colluvial deposits (Q-ca) at the east of the city and clays (Q1-l) at the central area of Bogotá (See Appendix B Fig. B1). Additionally, due to an excessive extraction of water from underlying aquifers, there is strong subsidence in the city, reaching values of 3.5 cm/year (Mora-Paez et al., 2020). Furthermore, Bogotá's rapid growth in the first half of the 20th century caused that nearly 35% of the houses in the city were built over 40 years ago (Departamento Administrativo Nacional de Estadística, 2018a).

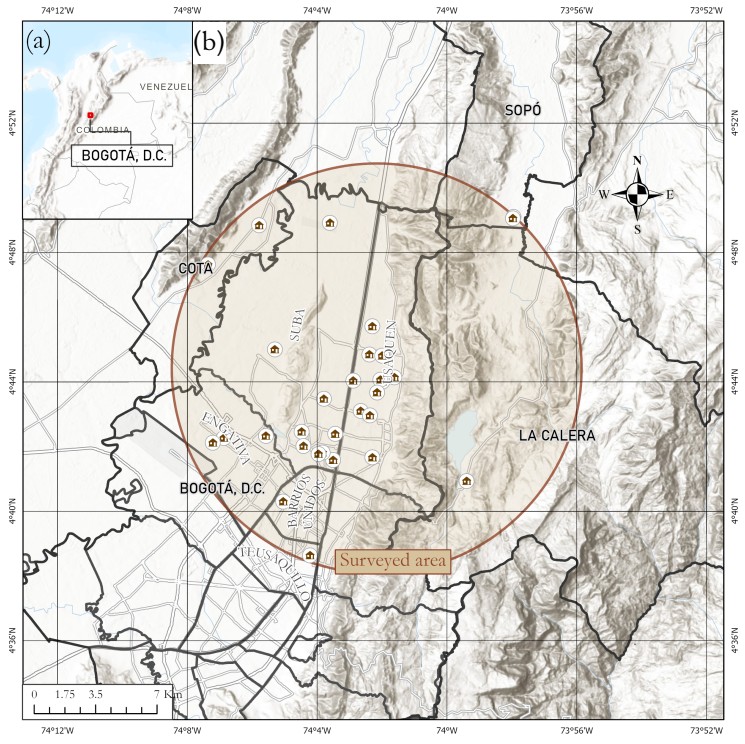

**Figure 2.** Map of the study area **(a)** General map of Colombia with the study area marked in red **(b)** Detailed map of the surveyed area with the 30 dwellings where measurements were made. Basemap source: Esri, HERE, Garmin, FAO, NOAA, USGS, © OpenStreetMap contributors, the GIS User Community and Departamento Administrativo Nacional de Estadística (2018b). Figure created by authors.

**2.1.2 222Rn sampling**

Since this study was performed during the COVID-19 pandemic, an online registration form was sent via social media and filled by 79 owners of residences who wanted to participate. Based on previous studies the selection criteria to determine



which dwellings would be sampled were: Type of residence (house or apartment), presence of basement (Lorenzo-González et al., 2017; Giraldo-Osorio et al., 2021; Li et al., 2022) and the lithology below the dwelling (Gundersen et al., 1993; Salazar et al., 2004; Maestre and Iribarren, 2018). The influence of these factors on IRC is presented on Table 1. The selection of houses was also made to ensure that indoor measurements in the 5 localities surveyed, would have a minimum sampling rate of 1 sample every 15 km$^2$.

**Table 1.** Criteria considered to select the studied residences based on factors reported to enhance indoor radon concentrations (Gundersen et al., 1993; Giraldo-Osorio et al., 2021; Li et al., 2022).

| Parameter | Reasons | Reference |
|---|---|---|
| *Type of residence* | Only houses were selected to ensure that the residence had at least one level in contact with the ground. Concentrations on ground floors usually are higher than upper levels. | (Lorenzo-González et al., 2017; Giraldo-Osorio et al., 2021) |
| *Presence of basement* | Six houses with basements were selected. The high density of radon (9.73 g/L) favours the accumulation of the gas on lower levels, such as basements. | (Lorenzo-González et al., 2017; Li et al., 2022) |
| *Lithology* | Houses with three different lithologies were included on the survey. Lithologies with higher contents of uranium have higher radon emanation rates. | (Gundersen et al., 1993) |

Several methods for measuring radon currently exist, however, the most economic and widely used is the alpha-track detectors (e.g. Field, 2015). In this study, Kodak LR-115 alpha-track detectors were placed in the selected houses for a period of 35 days between February 20 and March 28 of 2021. The detectors were supplied by the Nuclear Physics, Application and Simulation (FINUAS) laboratory, and the diffusion chambers were assembled as shown on Fig. 3.

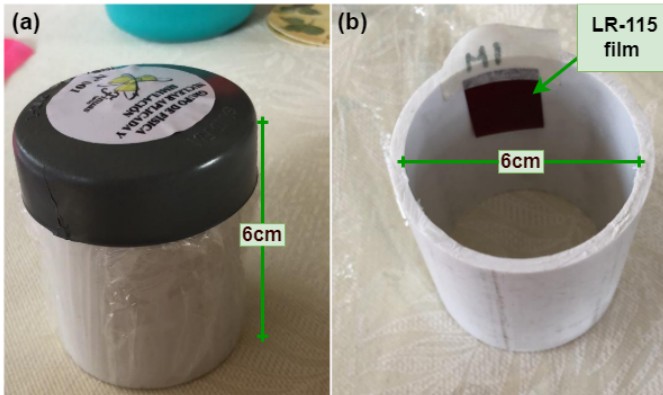

**Figure 3.** Images showing the diffusion chambers used to detect radon. **(a)** Outside of the diffusion chamber assembled. **(b)** Inside of the chamber with the location of the LR-115 film. Figure created by authors.

The LR-115 films were placed in the upper part of the diffusion chamber as described by Nikolaev and Ilić (1999) (See Fig. 3b). The dimensions of the diffusion chamber were selected to ensure that the conversion factor (1.4 tracks·cm$^{-2}$ / kBq·m$^3$·h)



used by Faisca et al. (1992) could be applied. This conversion factor has been used with LR-115 films and validated against
active methods (See Rojas-Arias et al., 2020, for details).

The alpha-track detectors were placed at a height of 60-180 cm above the ground, at a minimum distance of 15 cm from
the walls and away from doors, windows, and electronic devices to avoid environmental interference that may induce larger
measurement errors (Lorenzo-González et al., 2017). Additionally, a thin film of polyethylene was added to the lower part of
the chamber to avoid humidity and/or dust. Once installed, residents of the dwellings sent pictures of the sensors on a weekly
basis so that any changes in the location and /or conditions of the detector could be recorded.

After the 35 days exposure, the detectors were retrieved and shipped to the FINUAS lab in Tunja, Colombia, where they
were analysed. The analysis consisted of etching the LR-115 films with the process described by Rojas-Arias et al. (2020).
The alpha track density ($\rho_{tr}$) was then determined by counting the number of etched tracks per square centimetre under the
microscope. Afterwards, IRC was calculated using the conversion factor determined by (Faisca et al., 1992) as described on
equation 1.

$$IRC[Bq/m^3] = \frac{\rho_{tr}}{t \cdot CF \cdot 1000} \tag{1}$$

Where, $\rho_{tr}$ represents the density of alpha tracks (tracks·cm$^{-2}$), $t$ is the time exposed (h) and the conversion factor (CF = 1.4
tracks·cm$^{-2}$ / kBq·m$^3$·h) is the same determined by Faisca et al. (1992).

To evaluate the precision of the detectors, duplicate LR-115 films were included in 11 of the diffusion chambers selected
randomly and placed in front of the other film inside the diffusion chamber. The relative percent difference (RPD) was calcu-
lated as suggested by WHO (2009) (See Equation 2).

$$RPD = \frac{|IRC_{\text{sample}} - IRC_{\text{duplicate}}|}{mean(IRC)} \times 100 \tag{2}$$

### 2.1.3   Independent variables data acquisition and pre-processing

The predictor variables for fitting the regression model, which include geologic, meteorological, and building-specific factors,
were acquired from multiple sources. These sources included an online form completed by the participants and six other dis-
tinct data sources listed in Table A1. The spatial distribution of the meteorological and geologic variables can be found in
Appendix B on the supplementary material.

The regression model used three dummy variables which included the house age and two related to the lithology. For the



variables explaining the lithologies in the area, the reference category was selected to be the lithology characterized by clays (Q1-l). Furthermore, the houses built after 1980 were chosen as the reference category for the house age variable.

Finally, the dataset to which the regression model was applied was retrieved from Bogotá's cadaster dataset. A subset of the dataset was selected making sure that the residences followed two criteria: i) buildings with less than three stories (houses) ii) buildings where one of the three studied lithologies, analyzed in the statistical analysis, was present. The data of the predictor variables was finally added to each house in this dataset.

The result of the pre-processing resulted in two tabular datasets that from now on will be called: In situ dataset and cadaster dataset for simplification purposes. In the first one, the data consisted of the in situ measurements and the predictor variables. On the other hand, the second one contains the spatial and predictor information of the houses in the cadaster to which the regression model was applied.

## 2.2 Dashboard for indoor Rn modeling

Once both the radon measurements and the predictors data has been gathered, the IRC can be modeled by using the dashboard created in this study. In the following section, both the underlying statistical analysis and the dashboard architecture are presented.

### 2.2.1 Statistical analysis

The statistical analysis consisted of: i) fitting a log-linear regression model to the dataset with in situ measurements, ii) evaluating the performance using cross-validation, iii) undergoing a process of feature selection iv) estimating the radon concentration using the log-linear regression and v) visualizing the IRC estimations spatially (Figure 4).



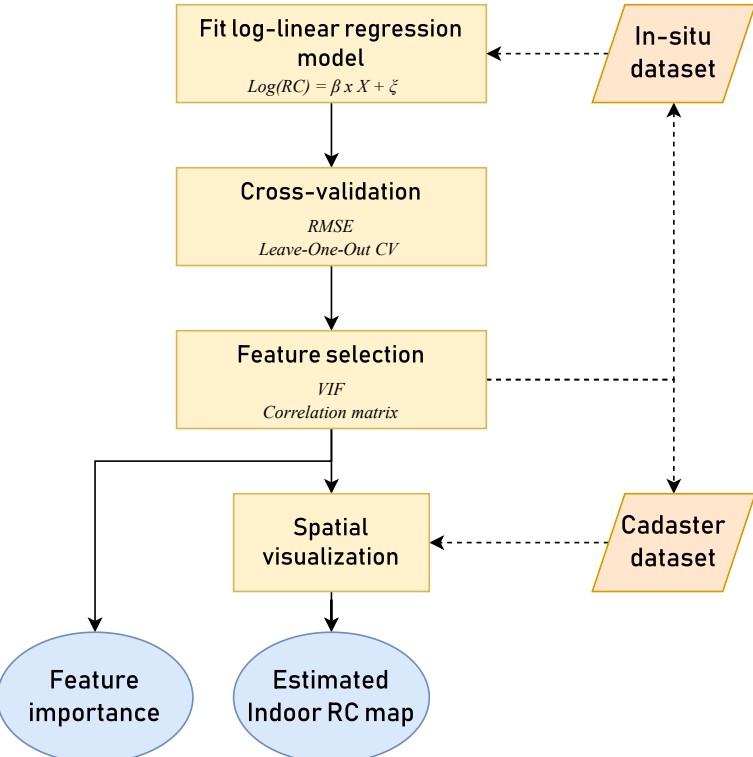

**Figure 4.** Flowchart of statistical analysis performed using ***Python 3.11***. Yellow colors represent processes executed for the statistical analysis. Blue boxes represent outputs of the analysis. Orange colors represent input datasets: The in situ dataset is composed of the in situ IRC measurements and the predictor variables. The cadaster dataset contains the spatial and predictor information of the houses in the cadaster to which the regression model was applied. Figure created by authors.

*Regression analysis*

A log-linear multivariate regression model was used to establish the association between one dependent variable ($\log(IRC)$)

and 5 independent variables. The model included geologic, meteorological and construction variables. Similar approaches have been used to estimate IRC in other study areas (Demoury et al., 2013; Vienneau et al., 2021; Alber et al., 2023). The log-linear regression model was fitted with equation 3. Every independent variable in the log-linear model can be interpreted by finding a percentage change in IRC due to an increase of that independent variable by a unit. This percentage change can be calculated for each variable as described in equation 4. All the statistical analysis was performed using the 'Statsmodels'and 'Sci-kit

Learn' packages in ***Python 3.11***.

$$\log(IRC) = \boldsymbol{\beta} \cdot X + \xi \tag{3}$$





Where, $\boldsymbol{\beta}$ represents a vector with coefficients for each independent variable, X a matrix with the values of the independent

variables and $\xi$ the error.

$$\text{Percentage change}_i \ [\%] = 100 \cdot (e^{\beta_i} - 1) \tag{4}$$

*Cross-validation*

The performance of the regression model was assessed using the leave one out cross-validation approach. This cross-validation
method gives an estimation of the test error that is approximately unbiased (Hastie et al., 2021). To measure the error of the
model, the root mean squared error (RMSE) was used as described in equation 5.

$$RMSE = \sqrt{\sum_{i=1}^{n} \frac{\left(\hat{y}_i - y_i\right)^2}{n}} \tag{5}$$

*Feature selection*

Originally, the log-linear model included four more variables that we considered could have an association with high IRC
(Fault proximity, temperature, urban/rural area and land subsidence). Nevertheless, they were later removed to reduce chances

of overfitting, and increase the model performance by focusing on the most informative features. According to their high
Variance Inflation Factor, land subsidence, temperature and urban/rural area variables introduced multi-co-linearity problems
to the model, therefore they were discarded. Moreover, since fault proximity values in houses of Bogotá are above the 150
meters threshold that previous authors have shown to have a significant association with higher IRC in dwellings, this variable
was not considered in the present study (Drolet and Martel, 2016). Removing these features did not substantially affect the

cross-validation estimation of the test error.

*Spatial visualization*

The log-linear multivariate regression model fitted with the in situ dataset was used to produce a map of estimated IRC with
the cadaster dataset. The IRC estimated for the houses was then aggregated in a grid of cells of size 100mx100m. The grid was

created and then visualised using the 'Geopandas' and 'Plotly' packages in ***Python 3.11***.

**2.2.2 Dashboard design**

A dashboard web application that allows the modeling and spatial visualization of estimated IRC by using in situ measure-
ments was developed. The dashboard layout consists of three main blocks that are shown in Fig. 5. In the first block, the user is



expected to add the two sets of tabular data (Previously called in situ and cadaster dataset). Then, the second block allows the
215 user to explore the in situ data, choose the parameters for the modeling and run the model (Left section of block 2 in Fig. 5).
The visualizations in this block show: The distribution of IRC measured with respect to reference levels, a correlation matrix
of the predictors in the in situ dataset and the variance inflation factor for the predicted variables (Right section of block 2 in
Fig. 5). Furthermore, The parameters that can be chosen by the user are: Type of model to be fitted, features selected to train
the model, high quality to get a higher spatial resolution result and the projected coordinate reference system in which the
220 cadaster data is projected. Finally, the third block exhibits the results of the model. On the left side the feature importance and
the cross-validation scores are presented, and on the right side the spatial distribution modeled is presented in an interactive
map.

The dashboard web application was built and deployed using the 'dash' package in **Python 3.11** and the servers of *pythonany-*
225 *where.com*. Furthermore, the dashboard can be accessed through this link: http://ircmodelingdashboard.eu.pythonanywhere.com/

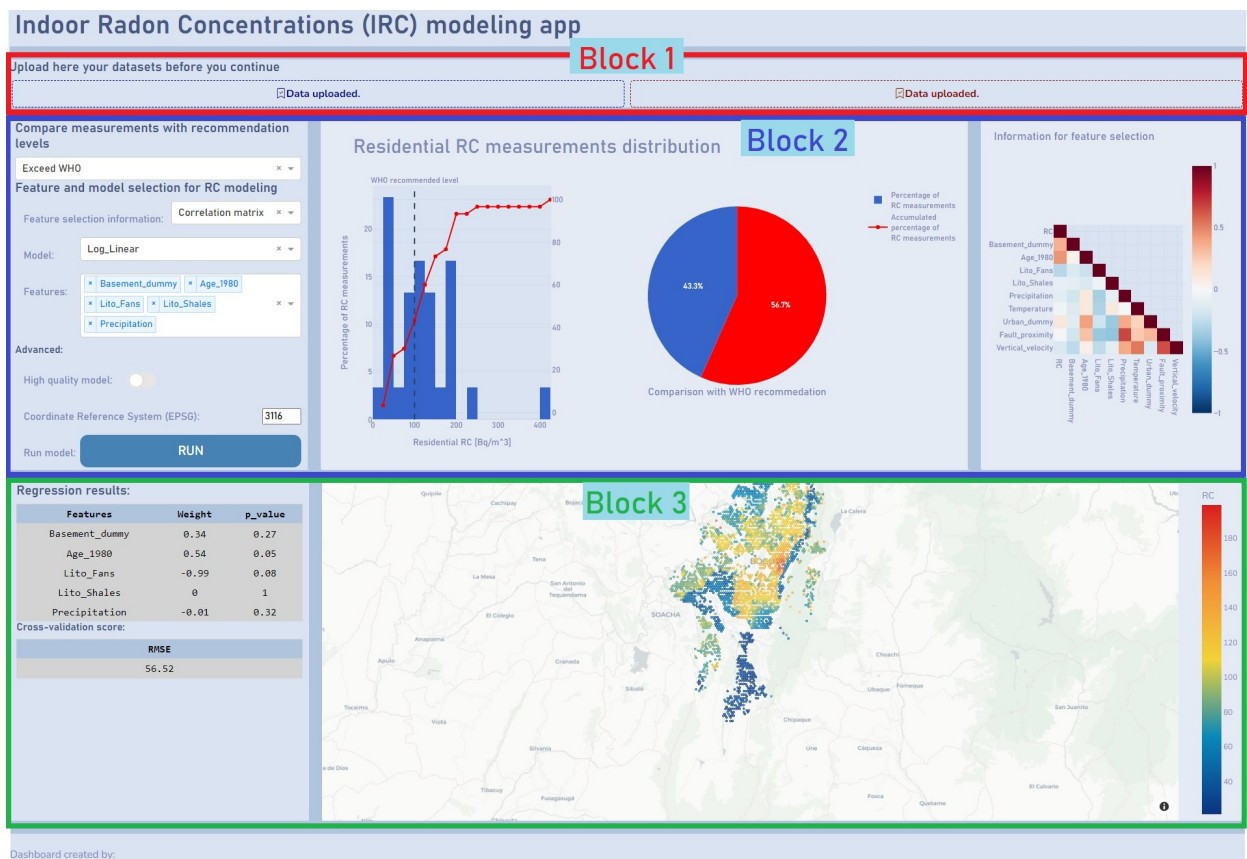

**Figure 5.** Overview of layout of the indoor radon concentration modeling dashboard developed. The three blocks that make up the dashboard
are highlighted in red (1), blue (2) and green (3). Block 1 is the data uploading block. Block 2 presents visualizations of the in situ data and
allows the user to chose the parameters and run the model. Block 3 is the results block. Figure created by authors.





# 3 Results and discussion

## 3.1 Indoor radon levels

The measured radon concentrations in the 30 dwellings selected ranged between 15 Bq/m$^3$ and 407 $\pm$ 10 Bq/m$^3$ with a geometric mean of 90.85 Bq/m$^3$ (Fig. 6a). The result of the 11 duplicates was a relative percent difference (RPD) of 14.05%

which falls within the typical uncertainty (10% - 25%) reported for alpha track detectors (WHO, 2009).

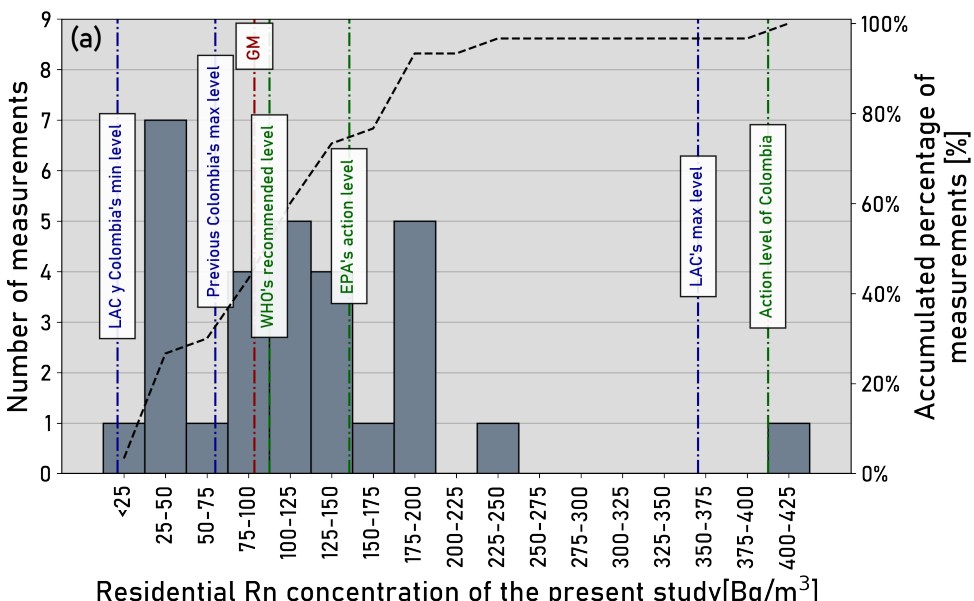

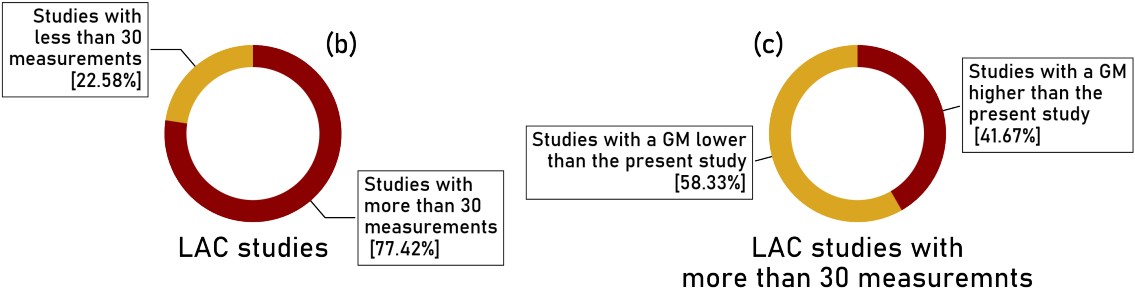

**Figure 6.** Measured radon concentrations shown as **(a)** Histogram of measured IRC in the Bogotá region within the context of Colombia and the Latin America and Caribbean (LAC) region. The geometric mean (GM) is indicated by the red dashed line. The black dashed line represents the accumulated percentage of samples. Green dashed lines represent reference levels and blue dashed lines show the minimum and maximum average values reported in studies in Colombia and the LAC region. **(b)** Ratio of studies made in LAC region with more or less than 30 measurements. **(c)** Ratio of studies with more than 30 measurements that have a greater or lower geometric mean than the one obtained in the current study. Information retrieved from Giraldo-Osorio et al. (2020). Figure created by authors.





### 3.1.1 Reported radon levels in LAC region and reference levels

The study of residential $^{222}$Rn in Latin America and the Caribbean (LAC) remains largely unexplored (Giraldo-Osorio et al., 2020). This was the third indoor $^{222}$Rn study conducted in Colombia and had the second largest number of observations (Giraldo-Osorio et al., 2020). Both previous studies were conducted in Manizales and presented mean IRC of 8.5 and 67.71

235  Bq/m$^3$ (Giraldo-Osorio et al., 2021). Our study presents the highest levels found in the country. The higher levels found in Bogotá with respect to Manizales could be explained by a smaller ventilation rate, related with a lower frequency of window and door opening, caused by lower mean temperatures (Chao et al., 1997). Additionally, the exclusive measurement in ground floors and basements in our study could also explain the high IRC levels measured (Lorenzo-González et al., 2017; Li et al., 2022). Due to its high density, $^{222}$Rn tends to accumulate in low-lying areas where ventilation is scarce.

The mean values reported in the 31 previous studies done in LAC ranged between 8.5 Bq/m$^3$ and 358 Bq/m$^3$ (Fig. 6a). The IRC measured in this study are in the upper range of the reported levels for the LAC region. Until 2020, it was reported that 24 out of the 31 studies had measured IRC in more than 30 residences (Giraldo-Osorio et al., 2020, Fig. 6b). The geometric mean IRC of the current study exceeds the geometric mean of 58.33% of these 24 studies (Giraldo-Osorio et al., 2020, Fig.

6c). This corroborates the high IRC measured in Bogotá compared to the values reported in the region.

The WHO's recommended level (100 Bq/m$^3$) was exceeded in 56.66 % of the residences analysed in this study (Fig. 7). Even though the Colombian action level (400 Bq/m$^3$) was exceeded only in one house (Fig. 7), there is enough scientific evidence to assert that the exposure to IRC, even below the WHO's recommended level, can affect human health (WHO, 2009).

Therefore, high IRC such as the ones found in this study raise concerns of the potential health impacts that $^{222}$Rn could be causing in Bogotá's population.



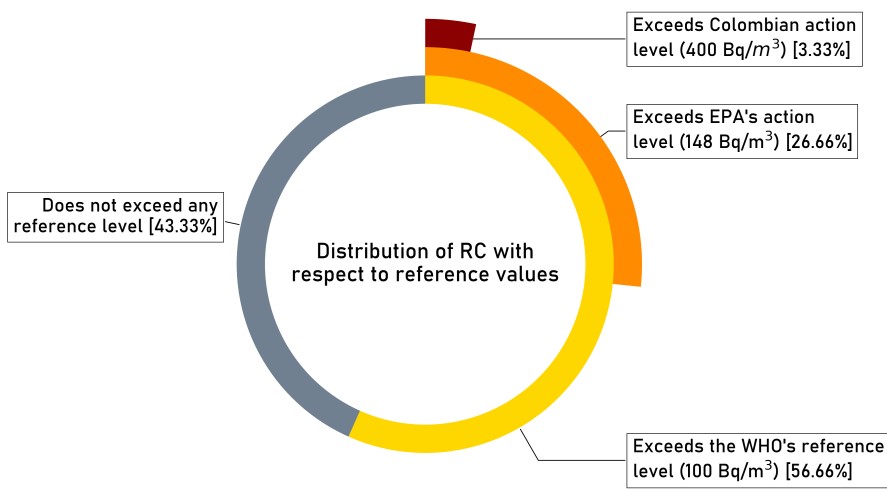

**Figure 7.** Pie chart with the percentage of the measurements exceeding the different reference levels. The reference levels considered were retrieved from WHO (2009), USEPA (1987) and Ministerio de Minas y Energía (2002). Figure created by authors.

## 3.2 Potential influential factors on IRC in the surveyed area

The statistical analysis performed considered 5 independent geologic, meteorological and construction variables. The results of feature importance derived from the dashboard are presented in Table 2. Moreover, the log-linear regression model presented a leave one out cross-validation RMSE of 56.52 $\frac{Bq}{m^3}$. The high RMSE reported can be explained by the small dataset used to train the model, it is expected that including more IRC measurements will substantially improve the model performance. Even though the estimated error is higher than the ones reported in previous studies, the log-linear model found significant associations with predictors that could explain high IRC values in the city (Vienneau et al., 2021). The inferential capability of the dashboard could therefore be used to focus future research and/or policy in high priority areas.





**Table 2.** Mean radon concentrations ($\overline{IRC}$) and results of the statistical analysis for each independent variable used as predictor. Results for the log-linear regression model ($\beta$ and $e^{\beta}$) are presented with a significance level represented by the p-value. $n$ represents the number of samples with that possible value.

| Variable | Type | Values | $n$ | $\overline{IRC}$ | $\beta$ | $e^{\beta}$ |
|---|---|---|---|---|---|---|
| *Basement* | Construction | | | | | |
| | | no | 23 | 100.34 Bq/m$^3$ | **0** | **1.000** |
| | | yes | 7 | 166.03 Bq/m$^3$ | 0.34 | 1.405 |
| *Age* | Construction | | | | | |
| | | Built after 1980 | 18 | 85.76 Bq/m$^3$ | **0** | **1.000** |
| | | Built before 1980 | 12 | 158.74 Bq/m$^3$ | 0.54** | 1.716 |
| *Lithologies* | Geologic | | | | | |
| | | (Q1-l) | 27 | 123.26 Bq/m$^3$ | **0** | **1.000** |
| | | (Q-ca) | 2 | 32.43 Bq/m$^3$ | -0.989* | 0.372 |
| | | (k1k6-Stm) | 1 | 77.20 Bq/m$^3$ | 0.0004 | 1.0004 |
| *Precipitation* | Meteorological | [71.54 - 137.18] mm | - | - | -0.0099 | 0.9901 |

\* = p-value<0.10
\*\* = p-value<0.05

House age was the only construction variable with a statistically significant ($p-value < 0.05$) association with IRC (Table 2). On the other hand, the association of geologic variables with IRC was only marginally significant for the dummy comparing alluvial fans and clays. Finally, the association of meteorological variables with IRC (i.e. precipitation) was not statistically significant in this survey.

Even though the influence of geologic variables on IRC has been extensively documented (e.g. Gundersen et al., 1993; Salazar et al., 2004; Maestre and Iribarren, 2018), the log-linear model fitted only showed a marginally significant association of lower IRC in houses that were built over alluvial fans and colluvial deposits (Q-ca) rather than over clays (Q1-l). The model suggests that IRC could be 63% lower in houses above alluvial fans and colluvial deposits compared to those above clays. This could be explained by the size of soil particles and the Uranium content in the different lithologies. Previous studies have reported positive correlations between Rn emanation and amount of soil particles with diameters below 0.1 mm (Thu et al., 2019), therefore higher Rn emanation rates could be present in clays rather than in alluvial and colluvial deposits that are mainly composed by a sandy matrix (Corredor et al., 2015). Furthermore, according to the Geochemical Atlas of Colombia the uranium content, which has been proven to determine the geogenic $^{222}$Rn potential (Gundersen et al., 1993), is slightly higher in soils containing clays (1.93 mg/kg) than in soils formed in alluvial fans and colluvial deposits (1.904 mg/kg) (Servicio Geológico Colombiano, 2016).

Several studies have concluded that residential IRC show important seasonal variations throughout the year because of changes in temperature and precipitation (e.g. Burke et al., 2010; Mullerova et al., 2017; Schubert et al., 2018; Crockett et al., 2018). The present study aimed to gauge the potential association of the spatial distribution of these two meteorological factors with IRC throughout a measuring period of 35 days. However, since both variables introduced multicolinearity problems to the re-



gression, the analysis was only done with precipitation data. Precipitation did not present a statistically significant association with IRC in the regression model. Despite this, previous studies suggest that the highest IRC in Bogotá could be measured in January and the lowest in May, due to the previously reported negative association of precipitation with IRC (Schubert et al., 2018; Mullerova et al., 2017; Grasty, 1994). Additionally, the results of previous surveys indicate that higher indoor $^{222}$Rn could be measured during night-time (5°C) in the city as a result of the negative association of mean air temperature with IRC

(Schubert et al., 2018). Temporal variations of IRC should be taken into account on future studies.

The main factor to correlate with IRC was found to be the house age ($r = 0.44$, $p-value = 0.01$, Appendix C Fig. C1). In addition, the log-linear model suggests that IRC in houses built before 1980, could be 71.6% higher than in houses built after 1980. The positive association of house age with IRC has been widely studied and is usually explained by the increase of

cracks and the lack of continuity of construction materials, which can enhance $^{222}$Rn migration (Karpińska et al., 2009). For the specific case of Bogotá's dwellings, those factors could be exacerbated throughout the years because of the occurrence of land subsidence throughout the city as it has been evidenced in cities with similar characteristics, such as Mexico City (Poreh et al., 2020). Likewise, land subsidence and the occurrence of other geologic phenomena such as earthquakes could, in turn, have a cumulative effect on structural damage of dwellings increasing IRC measured in old houses. Considering that 35.02%

of the houses registered in Bogotá's cadaster were built over 40 years ago, the positive association of this variable with IRC points to a high probability of having $^{222}$Rn hazard zones in the city.

Finally, a positive correlation was found between IRC and the measurement in a basement ($r = 0.35$, $p-value = 0.06$, Appendix C Fig. C1). This is supported by previous studies (e.g. Lorenzo-González et al., 2017; Giraldo-Osorio et al., 2021; Li

et al., 2022) and can be explained by the high density of the gas, which favours its accumulation in ground floors and basements (Field, 2015). Even though only 0.54% of the houses registered in Bogotá have a basement, measuring IRC in these areas was important since one third of the basements measured were used as bedrooms for domestic workers. This could increase significantly $^{222}$Rn exposure on this population group. Besides, most of the basements surveyed in the present studied were not registered in Bogotá's cadaster, thus cadaster's information could provide an underestimation of potential $^{222}$Rn exposure in

the city.

## 3.3  Potential hazard posed by residential $^{222}$Rn in Bogotá

The high IRC measured in the current study suggest that this gas could pose a health risk in the city. Likewise, events such as the COVID-19 pandemic could significantly increase $^{222}$Rn exposure in Bogotá's dwellings due to the implementation of measures like lockdowns, where residents are compelled to stay indoors. Given the size of Bogotá's population it is paramount

to conduct more sampling campaigns to understand the dimensions of the problem.



### 3.3.1 Prediction map

Considering the absence of monitoring programs in the country and that our study had a limited sample set, the modeling dashboard developed in this study was used to create a prediction map of the potential distribution of IRC. This map represents an alternative to assess the magnitude of $^{222}$Rn hazard in the city and establish target areas for future studies (Fig. 8). The map predicted IRC ranging from 21 to 220 Bq/m$^3$ and suggests that 32.09 % of the 551,570 houses in the model (177,008 residences) could have IRC above the WHO's recommended level (WHO, 2009, Fig. 8a). Considering that the mean household density in Bogotá is 2.9 inhabitants per household, a total of 513,323 people could be exposed to IRC above recommended levels. Notably, in the localities of Los Martires, Fontibón, Teusaquillo, Barrios Unidos and Puente Aranda average IRC were estimated to be above 100 Bq/m$^3$ (Fig. 8b). All of these localities are found near downtown Bogotá, which is the oldest area of the city. Even though the map created in our study shows the highest predicted IRC in zones built before 1980, it is necessary to extend the direct measurements to other areas to confirm our predictions.

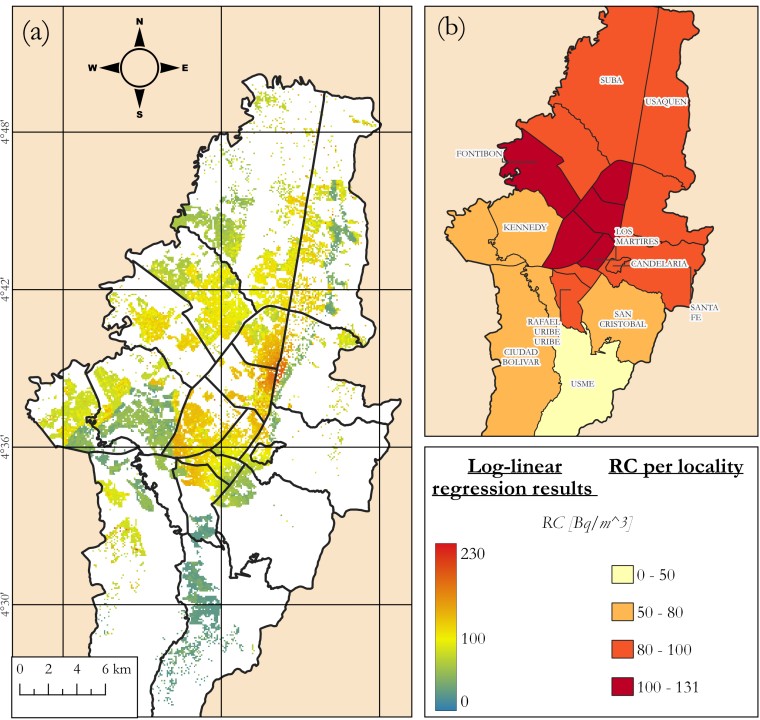

**Figure 8.** Application of log-linear regression model to houses registered on Bogotá's cadaster. **(a)** $^{222}$Rn concentrations predicted using the linear regression model fitted. Orange and red pixels exceed WHO's recommended level while green and blue values are below this level. **(b)** Predicted IRC in each locality of Bogotá. Basemap source: (Departamento Administrativo Nacional de Estadística, 2018b). Figure created by authors.



## 3.4 Dashboard performance

The dashboard built in this study was able to detect place-specific variables associated with residential IRC distribution in the city of Bogotá. In this case, a significant association between the house age and the residential IRC was found. This variable should be considered for the planning of further studies in the city. Moreover, the dashboard successfully provided summarized information of the in situ measurements, the effect of the variables on residential IRC modeled and the spatial distribution of IRC in Bogotá. These results show the great potential of the dashboard to assess this public health issue and guide national entities in the implementation of policies and monitoring programs to address this issue.

## 3.5 Study limitations

Even though the present study provides insight on what the situation of $^{222}$Rn hazard could be in the city of Bogotá, the findings must be evaluated with some limitations in mind. Firstly, the prediction map is built to have a better understanding of a potential distribution of IRC in Bogotá's dwellings, however, these estimations cannot be used to replace in situ IRC measurements. Secondly, to have a more accurate public health perspective in the analysis of indoor $^{222}$Rn, exposure to the gas should be quantified. Thirdly, in the statistical analysis most of the variables did not have a significant association with IRC and this could be due to the small sample set and not representative distribution. Finally, variations in IRC caused by meteorological factors should be determined considering the time variable and not only the spatial distribution. These limitations were the result of budgetary and time constraints. This preliminary study provides baseline information of IRC and indicates that more geological substrates as well as a temporally wider climatic range should be taken into account to improve the understanding of Rn distribution in the city.

## 4 Conclusions

High IRC were found in the Bogotá region with 56.66 % exceeding the WHO's recommended level and a geometric mean (90.85 Bq/m$^3$) that surpassed the mean values found in 58.33 % of the previous studies done in Latin America and the Caribbean. Furthermore, the high IRC measured points out to the importance of updating the residential Rn action levels in Colombia to address a potential public health hazard posed by Rn. These public policies should be accompanied by a rigorous monitoring of the exposure to the gas of people inside residences.

The house age and the presence of clays rather than alluvial and colluvial deposits, showed a significant and marginal association with high IRC respectively. The log-linear model fitted suggests that IRC in houses built before 1980 could be 71.60% higher than those after 1980. This could be due to the increase of cracks in the dwelling, a phenomenon that could be exacerbated by the occurrence of land subsidence in the city. The presence of alluvial and colluvial deposits rather than clays could decrease IRC as a consequence of the size of the soil particles and the different Uranium content in these lithologies.



A prediction map was built in the dashboard using a log-linear regression model to establish high Rn areas in the city. The map suggests that the oldest areas in Bogotá could present high radon concentrations. According to the estimations of this map, 513,323 citizens in Bogotá could be exposed to concentrations above WHO's recommended level. While it is a first approach, our prediction map cannot replace IRC measurements.

The dashboard built in this study joined with the use of inferential statistics demonstrated to be an efficient alternative for bridging the baseline information gap that is present in countries like Colombia, where IRC measurements are scarce. The performance of the dashboard was evaluated with the results of an exploratory study in Bogotá, Colombia.

While this is an important study to bridge the baseline $^{222}$Rn information gap in the city, considering that this study used only 30 measurements of IRC, it is important to expand the coverage of residential Rn monitoring campaigns in Bogotá, focusing particularly on the localities with older houses that according to the dashboard results could present a higher potential of having high IRC.

*Code and data availability.* https://github.com/mdominguezd/RnSurvey_Bogota_DataAnalysis



## Appendix A: Data acquisition

**Table A1.** Information of the data acquisition of the initial independent variables used as predictors in the log-linear regression model.

| Variable | Source | Reference | URL |
|---|---|---|---|
| *Basement* | Participants | Online form | - |
| *Age* | Participants & UAECD | Online form & (Unidad Administrativa Especial de Catastro Distrital, 2020) | https://mapas.bogota.gov.co/ |
| *Urban/Rural Area* | SDP | (Secretaría distrital de planeación, 2019) | datosabiertos.bogota.gov.co |
| *Lithology composed of alluvial fans and colluvial deposits (Q-ca).* | SGC | (Gómez Tapias et al., 2015) | sgc.gov.co |
| *Lithology composed of clays, peats and thin layers of gravels (Q1-l).* | SGC | (Gómez Tapias et al., 2015) | sgc.gov.co |
| *Lithology composed of shales, limestones, cherts and sandstones (k1k6-Stm).* | SGC | (Gómez Tapias et al., 2015) | sgc.gov.co |
| *Fault proximity* | SGC | (Gómez Tapias et al., 2015) | sgc.gov.co |
| *Vertical velocity (Absolute value)* | Sentinel 1B & ASF | - | search.asf.alaska.edu |
| *Precipitation* | IDEAM & RMCAB | - | rmcab.ambientebogota.gov.co visor.ideam.gov.co |
| *Temperature* | IDEAM & RMCAB | - | rmcab.ambientebogota.gov.co visor.ideam.gov.co |



# Appendix B: Spatial distribution of independent variables

## B1 Geologic variables

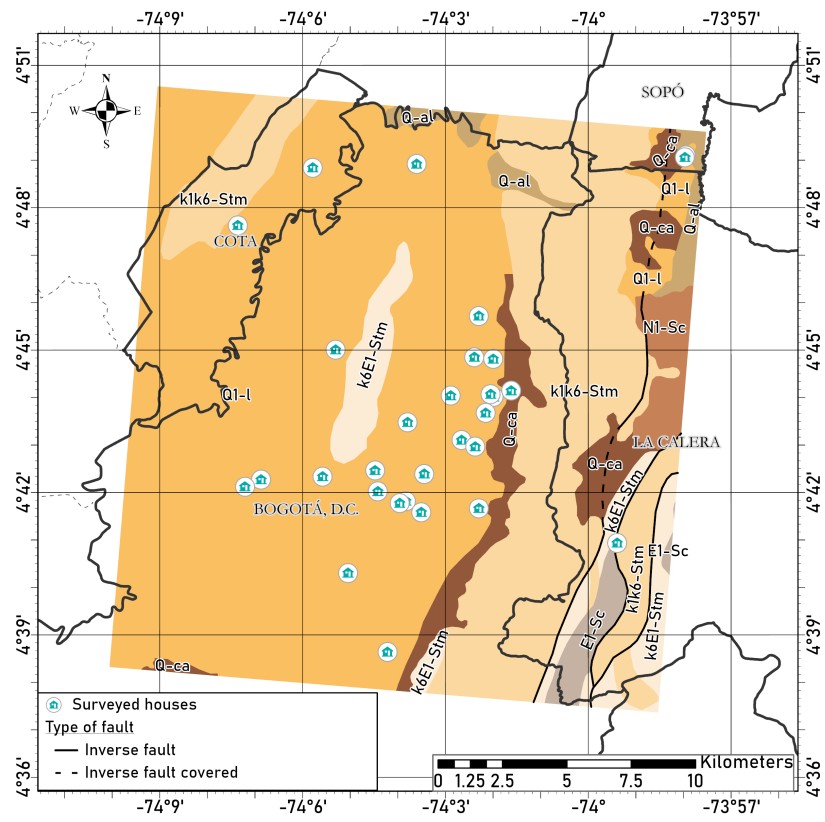

**Figure B1.** Lithology in the Bogotá region. Information retrieved from Gómez Tapias et al. (2015).



**B2 Meteorologic variables**

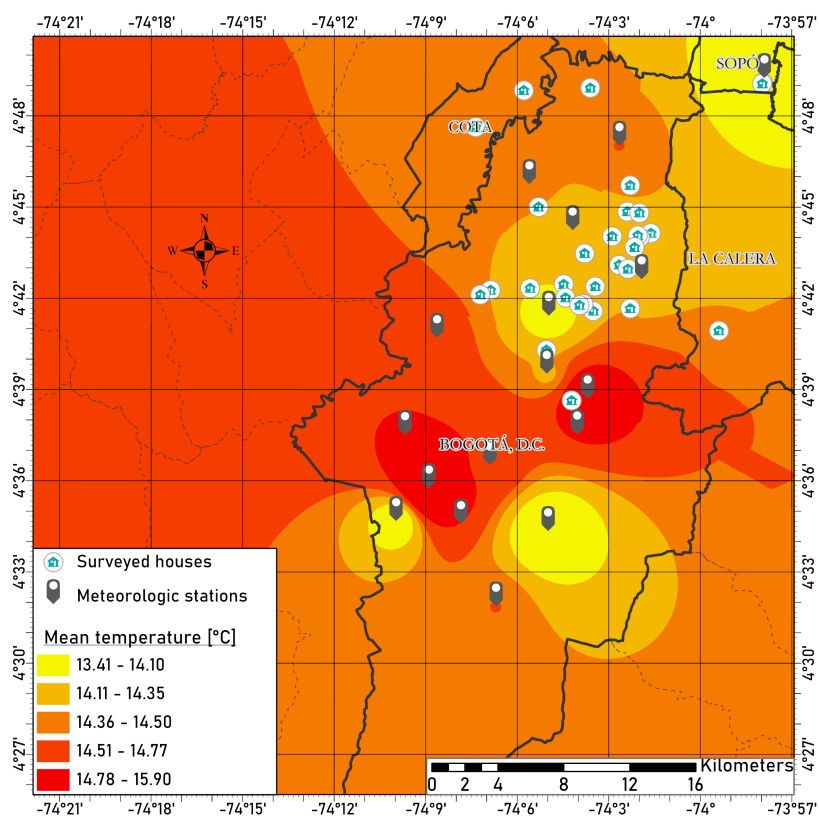

**Figure B2.** Mean temperature over the time window in the Bogotá region. Information retrieved from IDEAM and RMCAB and interpolated using Inverse Distance Weight.



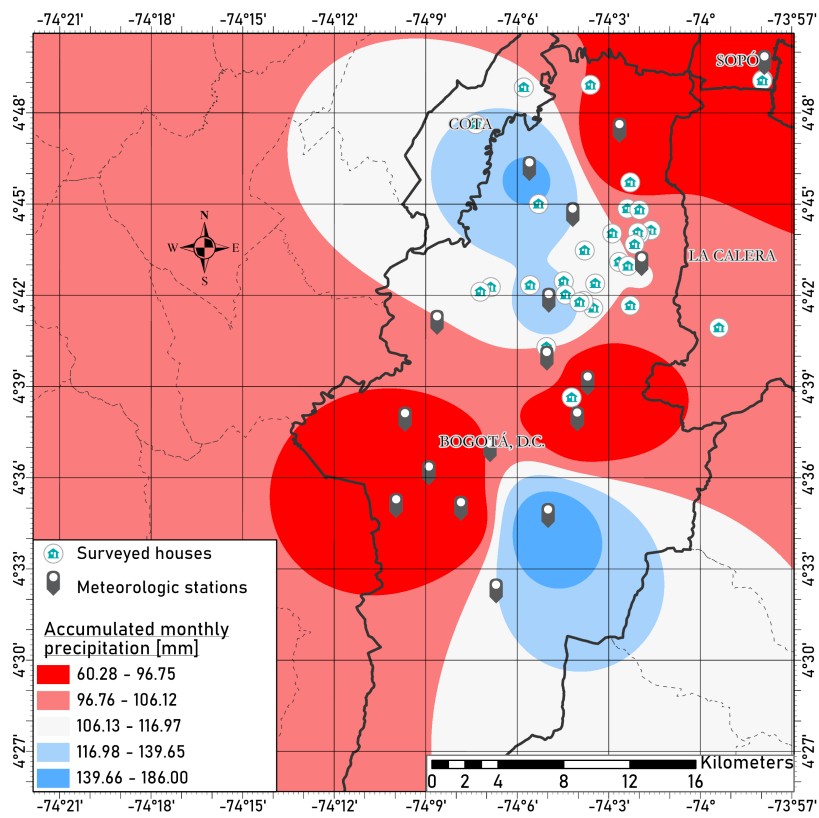

**Figure B3.** Accumulated monthly precipitation over the time window in the Bogotá region. Information retrieved from IDEAM and RMCAB and interpolated using Inverse Distance Weight.





**Appendix C:  Correlation coefficients**

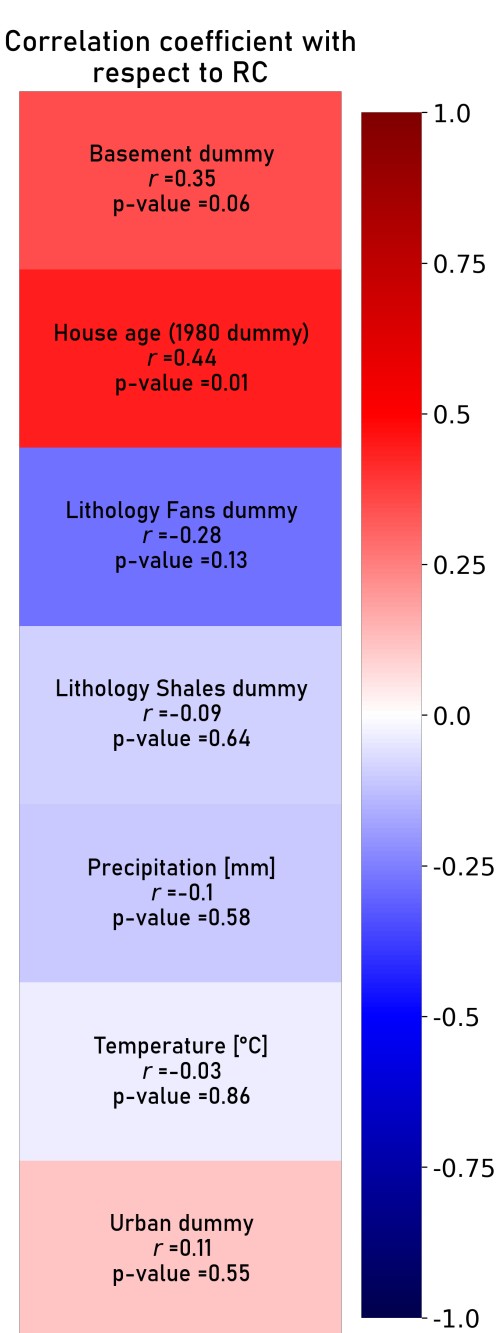

**Figure C1.** $r_{pearson}$ correlations with RC for independent variables studied (Table 2)



*Author contributions.* **MDD:** Conceptualization, Methodology, Software, Formal Analysis, Investigation, Data Curation, Writing - Original Draft, Visualization and Project Administration **MASG:** Methodology, Validation, Investigation, Resources, Data Curation, Writing - Review & Editing and Supervision. **CH:** Conceptualization, Investigation, Writing - Review & Editing, Supervision and Project Administration.

*Competing interests.* The authors declare that they have no conflict of interest

*Acknowledgements.* The authors express special gratitude to Professor Juan Pablo Ramos Bonilla for his orientation during the development of the statistical analysis and the manuscript writing. Additionally, the authors show gratitude to Professor Laszlo Sajo-Bohus for his support and orientation during the writing of the manuscript. The authors also thank FINUAS (Applied Nuclear Physics and Simulation) laboratory for their collaboration with the measurement reading and the donation of LR-115 films. The authors want to thank the Faculty of Science at
380 the Universidad de los Andes for providing funding through project - INV-2021-128-2286 to Carme Huguet.



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
