# Peer review of "Indoor 222Rn Modeling in Data-Scarce Regions: An Interactive Dashboard Approach for Bogotá, Colombia"

_EGUsphere, 2023_

## Author Response (AR1)

Dear editor and reviewers,

We are very grateful for your constructive feedback and thoughtful suggestions, which have enormously helped enhancing the quality of our manuscript. As you can see on the revised manuscript, all your suggestions have been considered and incorporated.

Below, you will find your suggested revisions along with our responses, outlining how each suggestion has been incorporated into the revised manuscript.

Once again, thank you for the dedicated time.

All the best,

Martín, Carme and María Angélica
* * *
**EDITOR**

- Figure 2 needs some improvement. All names/coordinates indicated must be readable. Find extra info about the figure guideline here: https://www.natural-hazards-and-earth-system-sciences.net/submission.html#figurestables
- Figure5. Idem as figure 2. However, here I may understand that the overall idea is to give an overview, and that not everything presented in the figure should be readable in details. Nevertheless, I propose that you provide in the caption a bit more description about the content of the figure to overcome this readability drawback.
- Figure 8. Geographical coordinates are not readable. Increase the font size.
- Figure B2 Explain the acronyms in the caption.
- Figure B3. Explain the acronyms in the caption. The legend of the precipitation should use rounded as I think, unless I am wrong, that decimals do not make sense here.

All figures stated above have been changed considering the editor's suggestions.

REV. 1

We appreciate the insightful comments provided by referee #1, which will undoubtedly contribute to the improvement of the manuscript. Below, we have addressed each comment individually:

- *I am not sure how dashboard is different in terms of objectives from other prediction strategies such as machine learning, geospatial approaches, mathematical models etc. Authors need to discuss other studies in detail for prediction of radon concentration in the direction of mapping. They should clearly bring the similarities or significant difference, if any for their study.*

We recognize the importance of clarifying the distinctive features of our dashboard in comparison to other prediction strategies. In our study, the interactive dashboard serves as a unique tool with specific objectives that allow us to reach a wider audience compared to other traditional approaches.

The primary objectives of our dashboard are:

1. To identify and comprehend the variables influencing radon distribution in a particular area.

2. To estimate IRC spatial distribution for a specific study area, which can help guide policy making and the direction of future studies.

3. To provide an accessible tool. The dashboard allows anyone with a correctly gathered dataset to estimate indoor radon concentration for their specific location without requiring an in-depth understanding of the underlying mathematical models.

While various prediction strategies, including machine learning and geospatial approaches, have been employed in similar studies, our dashboard stands out by combining mathematical models with a user-friendly interface that facilitates practical use without extensive technical expertise. This democratization of radon concentration estimation aligns with our goal of overcoming data scarcity challenges in regions where traditional studies might be limited.

This information will be included in the revised manuscript.

> • *Discussion on the mapping in different regions should be included in the introduction part after a careful literature survey.*

We agree that a more explicit discussion on the mapping of indoor radon concentration in different regions could enhance the introduction of our paper. While in our introduction we mainly address the lack of indoor radon measurements and the absence of national radon databases in the global south, we acknowledge that including information about mapping IRC will make the introduction more complete.

While we have a length constrain by the journal, we will integrate a concise discussion on the challenges and gaps in mapping indoor radon concentration, particularly in regions with limited data, such as the global south, since this is the scope of the study.

> • *Authors have highlighted the lack of baseline studies on radon concentration in Latin America and Caribbean (LAC) regions in introduction part. Subsequently they refer to LAC studies in fig. 6 (b) and 6 (c). How many such studies? Why*

*do they want to emphasize on the variability of GM in different studies? Is it expected or not expected?*

We appreciate the reviewer's inquiry and would like to provide clarity on the points raised. In our study, we referred to the lack of baseline studies on radon concentration in Latin America and the Caribbean (LAC) regions in the introduction. The specific studies conducted in this region are detailed in Giraldo-Osorio et al. (2020), which we cited in our manuscript.

While the exact number of studies is not explicitly mentioned in our paper, readers can find a comprehensive list and detailed information on Table 1 of the paper by Giraldo-Osorio et al. (2020). This reference serves as a valuable resource for those interested in exploring the existing studies on radon concentration in the LAC regions. The reference has been added here for the reviewer convenience.

*Giraldo-Osorio A, Ruano-Ravina A, Varela-Lema L, Barros-Dios JM, Pérez-Ríos M. Residential Radon in Central and South America: A Systematic Review. International Journal of Environmental Research and Public Health. 2020; 17(12):4550. https://doi.org/10.3390/ijerph17124550*

Regarding the inclusion of Geometric Means (GMs) from other studies in Figure 6, our objective is not to emphasize the variability of GM itself. Instead, we aim to provide a comparative context for our study's results in relation to existing studies in the LAC region. This approach helps readers understand how our findings align with or differ from the broader body of research in the same geographical context.

- *Why should single measurement leading to radon concentration > 400 Bq/m³, not be treated as outlier for the dataset? Generally, when the readings are exceedingly large for passive measurements, an active measurement check works out as a good support for ensuring the correctness of the estimate.*

We acknowledge the concern regarding the single measurement leading to a radon concentration exceeding 400 Bq/m³. In our study, we opted not to treat this value as an outlier due to the inherent log-normal distribution observed in indoor radon concentrations (IRC). The log-normal distribution of IRC explains the occurrence of relatively high concentrations in some instances. This variability is a characteristic feature of radon levels in indoor environments. Additionally, the value above 400 Bq/m3, while high, was not unexpected given that it is supported by several of the factors that have been previously linked to high radon values (Age of the house & measurement in basement).

Moreover, active measurements, which could serve as additional validation checks, were not available for us at the time of our study. We understand this limitation, and

we appreciate the reviewer's suggestion. In the revised manuscript, we will explicitly address this limitation in the study limitations section, highlighting the absence of active measurements and recommending their inclusion in future studies for enhanced accuracy and robustness.

> - *Authors should make fig. 8 in a better representative way. Measurement nodes can be inserted in the prediction map itself. Instead of showing block wise results in fig. 8 (b) (which may not be a proper way to gauge the average level for eastern regions), a colour coded flag representing the averaged value for the locality can be shown.*

We appreciate the reviewer's insightful suggestion regarding the representation of Fig. 8. We agree that improving the representation can enhance the clarity of the information presented. In the revised manuscript, we will add the location where measurements were taken in Fig 8a.

Nevertheless, we believe that the block wise representation in fig. 8b clearly shows how the estimations of indoor radon concentrations differ per locality, and the joint presentation with fig. 8a allows the reader to identify where the residences modelled are for each locality.

> - *In any case, prediction of concentration for locality which is a function of prediction from regression model again depends on the number of measurements. For a field environment, consisting of many variables, the data of measurement is still insufficient when mapping it to the entire region. The approach may lead to wrong estimate (due to insufficiency of input) and may then affect the policy considerations.*

We acknowledge the inherent challenges in mapping radon concentration across an entire region, particularly in a field environment with numerous variables and scarce measurements. Due to this, we included it in the study limitations section in the discussion.

As highlighted in the study limitations section (lines 331-335), we recognize that the prediction map serves the purpose of providing insights into the potential distribution of Indoor Radon Concentration (IRC) in Bogotá's dwellings. However, it is crucial to emphasize that these estimations do not serve as a replacement of in situ IRC measurements. The map is a tool for better understanding and guiding further studies rather than a substitute for direct measurements.

Moreover, we acknowledge that the statistical analysis has limitations which may be attributed to the relatively small sample size and potentially non-representative

distribution. This reinforces the importance of expanding studies to gather more comprehensive datasets for a more robust analysis.

- *How do we validate prediction map? It may be a better strategy to leave out 10-20 % of the input data and then validate the same after prediction. Validation for random localities is an important factor in my view; authors must devise an intuitive plan to tackle this issue.*

In our study, we employed the leave-one-out cross-validation approach to assess the performance of the regression model (And prediction map). This method was chosen due to the relatively small size of our dataset and the fact that this method leads to the least biased estimation of the test error (Lines 190-192).

However, we recognize the importance of considering alternative and possibly more robust validation strategies. For instance, the use of spatial cross-validation could offer additional insights into the validity of our map.

In the revised manuscript, we would like to explore the implementation of a spatial cross-validation approach to enhance the evaluation of our prediction map.

- *Authors should discuss findings and implications used in section 3.2 in better ways. This section should be re-written.*

We acknowledge and value your input. However, the term "better ways" is a bit broad, and we want to ensure that we address your specific concerns thoroughly.

Could you kindly provide more details or specific aspects you believe require improvement in Section 3.2?

**REV. 2**

We appreciate referee #2's thorough review and their enthusiasm regarding the publication of our study. We are certain that their suggestions will help enhance our manuscript. Below we answer each of their specific comment:

- *please consider limiting the number of significant digits throughout the manuscript, i.e. do not give decimal digits for Bq/m³ and %*

We acknowledge the need to present results more clearly. We will revise the manuscript to limit the number of significant digits, especially for values related to Bq/m³ and percentages.

- *clarify the model building procedure with respect to feature selection. At which stage was feature selection done? Currently it reads (L170ff) as if it was done after fitting the final model. Also Fig. 4 creates some confusion in this respect.*

We acknowledge that the model building procedure may lack clarity, and we recognize that Figure 4 might be contributing to this confusion. To address this concern, we will make specific adjustments to the figure by incorporating the fitting of the final model after the feature selection step. Moreover, we will amend the manuscript text by introducing a distinct step after feature selection and before estimating radon concentrations (line 172) to explicitly communicate the sequencing of these processes. These modifications will help provide a more transparent representation of our model building approach.

- *L239: accumulation of radon in basements is a consequence of its proximity to the source. The specific (high) weight of the radon atom itself plays no fundamental role.*

Thank you for pointing this out. We will modify the language in line 239 to emphasize that radon accumulation in basements is primarily due to its proximity to the source and not to its atomic weight.

- *L256: RMSE is usually quite high in radon modelling. I agree that a larger sample size would probably improve the model performance, but still a significant prediction uncertainty will remain most likely due to generally imperfectness of environmental predictor data and absence of some crucial information (e.g., airtightness of buildings, ventilation intensity of residents)*

We acknowledge the significant challenges associated with the modelling of indoor radon concentrations. To improve the discussion of section 3.2 we will update L256 to highlight that even though a larger sample size might improve the model performance, a significant prediction uncertainty is associated to radon modelling due to the imperfect environmental predictors and/or the absence of certain crucial information.

- *section 3.3.1 prediction map: a reliable estimation of the number of people exposed to concentrations above a certain threshold (100 Bq/m³ in this case) requires consideration of prediction uncertainty. The results of the log-linear regression represent the conditional mean. However, even if the predicted conditional mean lies below the threshold there is still the chance that a significant fraction of households with the same predictors/characteristics exceed the threshold. This is a consequence of prediction uncertainty (see L255). Thus, the fraction of households exceeding the threshold, while the conditional mean is below the threshold, increases with increasing prediction uncertainty. Hence, I recommend to either predict the full conditional*

*distribution for estimation of exceedances above a threshold or modifying the interpretation of the results and stating that the estimated numbers rather reflect minimum numbers (due prediction uncertainty and lognormality of the distribution).*

We highly appreciate your insightful recommendation. In response, we intend to address this concern using the second approach you proposed. Specifically, we will modify the interpretation of the results to explicitly state that the estimated numbers should be regarded as minimum values. This adjustment will adequately account for prediction uncertainty and will offer a more transparent and comprehensive perspective on the exposure estimates.

---

## Author Response (AR2)

Dear editor,

We are very grateful for your constructive feedback and thoughtful suggestions, which have enormously helped enhancing the quality of our manuscript. As you can see on the corrected manuscript, all your suggestions have been considered and incorporated.

Once again, thank you for the dedicated time.

All the best,

 Martín, Carme and María Angélica